# A Quality Improvement Project to Support Post-Intensive Care Unit Patients with COVID-19: Structured Telephone Support

**DOI:** 10.3390/ijerph19159689

**Published:** 2022-08-06

**Authors:** Sabine A. J. J. op ‘t Hoog, Anne M. Eskes, Jos A. H. van Oers, José L. Boerrigter, Meike W. J. C. Prins-Smulders, Margo Oomen, Johannes G. van der Hoeven, Hester Vermeulen, Lilian C. M. Vloet

**Affiliations:** 1Department of Intensive Care, Elisabeth-Tweesteden Hospital, 5022 GC Tilburg, The Netherlands; 2Research Department of Emergency and Critical Care, HAN University of Applied Science, 6525 EN Nijmegen, The Netherlands; 3Department of Surgery, Amerstam UMC, University of Amsterdam, 1105 AZ Amsterdam, The Netherlands; 4Menzies Health Institute Queensland, School of Nursing and Midwifery, Griffith University, Gold Coast, QLD 4222, Australia; 5Department of Intensive Care Medicine, Radboud University Medical Centre, 6525 GA Nijmegen, The Netherlands; 6Radboud University Medical Centre, Radboud Institute for Health Sciences IQ Healthcare, 6500 HB Nijmegen, The Netherlands; 7Foundation Family and Patient Centered Intensive Care, 1801 GB Alkmaar, The Netherlands

**Keywords:** intensive care unit, family-centered care, nurse-led, evidence-based quality improvements, COVID-19

## Abstract

Background: More than 50% of intensive care unit (ICU) survivors suffer from long-lasting physical, psychosocial, and cognitive health impairments, also called “post-intensive care syndrome” (PICS). Intensive care admission during the COVID-19 pandemic was especially uncertain and stressful, both for patients and for their family. An additional risk of developing symptoms of PICS was feared in the absence of structural aftercare for the patient and family shortly after discharge from the hospital. The purpose of this quality improvement study was to identify PICS symptoms and to support post-intensive care patients and families in the transition from the hospital to the home. Therefore, we offered post-ICU patients and families structured telephone support (STS). Methods: This was a quality improvement study during the 2019 COVID-19 pandemic. A project team developed and implemented a tool to structure telephone calls to identify and order symptoms according to the PICS framework and to give individual support based on this information. We supported post-ICU patients diagnosed with COVID-19 pneumonia and their family caregivers within four weeks after hospital discharge. The reported findings were both quantitative and qualitative. Results: Forty-six post-ICU patients received structured telephone support and reported symptoms in at least one of the three domains of the PICS framework. More than half of the patients experienced a loss of strength or condition and fatigue. Cognitive and psychological impairments were reported less frequently. Family caregivers reported fewer impairments concerning fatigue and sleeping problems and expressed a need for a continuity of care. Based on the obtained information, the ICU nurse practitioners were able to check if individual care plans were optimal and clear and, if indicated, initiated disciplines to optimize further follow-up. Conclusions: The implementation of the STS tool gave insight in the impairments of post-ICU patients. Surprisingly, family caregivers expressed fewer impairments. Giving support early after hospital discharge in a structured way may contribute to providing guidance in the individual care plans and treatment of the early symptoms of PICS (-F).

## 1. Introduction

Since March 2020, the world has faced the outbreak of COVID-19 pneumonia, which has led to enormous pressure on the healthcare system [1]. Although the majority of individuals infected with COVID-19 develop mild symptoms and recover without hospitalization, some patients require intensive critical care [2].

In the Netherlands, the first COVID-19 infection was confirmed on 27 February 2020. Since the outbreak, more than 13,550 patients with a COVID-19 infection have been admitted to the ICU [3,4]. Currently, 20% of all Dutch ICU beds are occupied by COVID-19 patients. On average, six ICU patients per day were admitted with COVID-19 in July 2022 [5]. At the peak of the first “wave”, hospitals gradually increased the number of ICU beds from 1150 (6.4 beds per 100,000 citizens) to 1700 during March and April 2020. The COVID-19 pandemic has directly affected local ICU hospital policies, including isolated nursing, increased visitor restrictions, and time restraints to give appropriate information to patients and their families [6,7]. Furthermore, patients with a COVID-19 infection can rapidly deteriorate to complete respiratory failure with severe consequences and a high risk of mortality [8]. This uncertainty of the progress of the disease and changing hospital policies may negatively influence the newly experienced health burden after ICU discharge for patients and family caregivers already at risk of developing post-intensive care syndrome (PICS) [9] and post-intensive care syndrome—family (PICS-F) [9]. 

During the COVID-19 pandemic, no structured aftercare was organized for post-ICU patients [10]. There is little evidence on the effectiveness of interventions to reduce the impact of PICS (-F) during the transition after ICU admission and follow-up care [11,12]. However, early follow-up during the transition may help to detect patients and families at risk for PICS early; improve the information provision to patients, families, and other healthcare professionals; and start early interventions [11]. Currently, COVID-19 patients occupy 20% of all Dutch ICU beds. On average, six ICU patients per day were admitted with COVID-19 in July 2022 [5]. Due to the COVID-19 pandemic, access to healthcare and communication strategies are still limited. As a result, patients, family members, and healthcare workers experienced psychological symptoms such as stress, anxiety, depression, and fear, which may lead to burnout [13,14]. Virtual patient- and family-centered communication is recommended to improve meaningful communication between healthcare workers and patients and their family members [15]. Telehealth, including video or audio communication such as structured telephone support (STS), has been fully embraced as an intervention to keep the patient and family involved [16,17]. Several studies describe the positive impact of telehealth, which may reduce the impact of social isolation [18,19,20]. Negro et al. (2020) describes a structured intervention by video-calls in ICUs for patients and their family members [21]. Initiating communication between patients and close family members can improve the emotional experience and potentially reduce the psychological burden. In addition, several studies describe that meeting patients’ and family members’ needs also may reduce the stress systems of professionals [22,23,24].

STS includes an individual assessment of the patient’s and/or family’s needs, which may be an appropriate intervention in the context of a persistent pandemic [25]. Based on previous research in heart failure patients, it has been shown that STS improves patient outcomes by reducing mortality and readmission rates [26,27]. It also results in higher patient satisfaction rates, better care experiences, fewer post-discharge problems, enhanced self-management, and lower costs [28,29,30,31].

A recent study showed that 90% of the post-ICU patients with COVID-19 who received STS follow-up reported symptoms within at least one PICS domain [32]. At 1 month after discharge, more than one-third of patients reported acute stress disorders or cognitive impairments [32]. Within 3 and 6 months after discharge, 90% of patients still reported symptoms within one PICS domain and reduced health-related quality of life (HRQoL) [33,34,35]. Even after one year of survival after critical COVID-19, survivors reported frequent symptoms within at least one PICS domain [36]. 

The primary aim of this quality improvement project was to provide support to post-intensive care patients and families; therefore, we developed and implemented an STS tool for post-intensive care patients and families in the transition from the hospital to the home. The STS intervention was undertaken by ICU nurse practitioners and underpinned by the PICS framework [9]. It comprised several short questions focusing on the different post-ICU problems patients can experience with PICS (i.e., physical, mental, and cognitive consequences). Second, to provide a deeper understanding of the long-term impairments of post-ICU patients and their families, we report the findings of the STS in the PICS framework [9].

## 2. Materials and Methods

In this quality improvement study, post-ICU patients treated for COVID-19 pneumonia received an STS within four weeks post-discharge to home. We used both quantitative and qualitative methods integrated in the Deming cycle, as reported by the criteria of the revised Standards for Quality Improvement Reporting Excellence 2.0 framework [37].

### 2.1. Study Design and Setting

This was a quality improvement initiative during the COVID-19 pandemic in a Dutch tertiary hospital. This hospital has a 34-bed, mixed medical/(neuro) surgical ICU with a 24/7 intensive care service delivered by ICU nurses and intensivists [38]. During the outbreak of COVID-19, the ICU capacity in this hospital was upgraded to 40 beds to guarantee care for both COVID-19 patients and regular ICU patients.

### 2.2. Population and Sampling

We offered STS to all post-ICU patients diagnosed with COVID-19 pneumonia, as proven by a reverse transcription–polymerase chain reaction (RT-PCR) test, who were discharged home in the period of 7 March 2020 to 15 May 2021. If family members were involved, they were also asked to participate.

### 2.3. Description of the Intervention

#### 2.3.1. Plan

At the start of the COVID-19 pandemic, the project team, including three ICU nurse practitioners (SH, MO, MP), an ICU physician, and nurse researchers, rapidly developed an STS tool—the so-called “STS Post-ICU COVID-19 tool”. The purpose of developing and implementing the tool was to structure a telephone support intervention to screen the symptoms experienced by post-ICU patients, divided into the three domains of the PICS framework. By asking about all the domains of PICS in a structured way, we tried to identify where the most symptoms and thus the greatest needs are in their individual treatment plan. During the phone interview, all symptoms were asked about in a structured way and noted using the STS Post-ICU COVID-19 tool. Then, the ICU nurse practitioner discussed with the patient and/or family whether there was sufficient support at home or whether additional support was required. Based on a rapid search of the literature from Medline/PubMed, focusing on systematic reviews, we identified evidence of the effectiveness of the use of STS intervention on patient outcomes [26,27]. Consequently, we combined the well-known PICS framework [9,39] and local knowledge of our team to define the content of the STS intervention for our population. 

Figure A1, Appendix B visualizes the final developed STS Post-ICU COVID-19 tool, which is a structured digital pocket card that focuses on the three domains of the PICS framework (physical, cognitive, and psychological function) [9]. In addition to this, we added one question about ICU experiences and two questions (5a and 5b, Figure A1, Appendix B) to assess the mental health and symptoms of the family caregiver. 

#### 2.3.2. Do

After development, we implemented the STS Post-ICU COVID-19 tool into daily practice. To support the use of the tool, we developed smart phrases (i.e., blocks of text that can automatically be pasted into the patient’s Electronic Health Record (EHR)). 

Three ICU nurse practitioners (SH, MO, and MP) conducted the intervention within four weeks after the patients’ hospital discharge to home. Due to the rapid implementation strategy due to COVID-19, the ICU nurse practitioners were not trained in the telephone support. Therefore, the ICU nurse practitioners discussed a strategy for the telephone call beforehand, and the tool provided sufficient structure for the telephone support to have a uniform procedure. All post-intensive care patients who were discharged from hospital to home were offered an STS within four weeks after hospital discharge.

Each STS call started with a short review of the patient’s experiences in the ICU. All questions, per the PICS domain, were asked using the structured format of the STS intervention. The occurrence of PICS symptoms per domain was assessed by patients using a four-point scale ranging from “no burden” to “very much”. In addition to this score, patients were asked to provide additional information to concretize the symptoms and report-initiated interventions or advice. This information was also reported in the EHR. Lastly, if family members were involved, they were asked about their psychosocial symptoms. If not, then patients were asked to assess the symptoms of their family members or next of kin. Each call lasted 20–60 min.

#### 2.3.3. Check

We described the gathered symptoms that patients and family caregivers reported during the STS. Therefore, we collected demographical data including age, gender, body mass index (BMI), and pre-existing comorbidities. Furthermore, we used the Acute Physiology and Chronic Health Evaluation IV (APACHE IV) to estimate the risk of short-term mortality and the length of the ICU stay [40]. The Sequential Organ Failure Assessment [41] was used to describe the severity of organ failure during ICU admission. We recorded the number of mechanical ventilation days, the length of stay (LOS) in the ICU, and the LOS in the hospital. All clinical data were retrospectively collected from the EHR. The data from the STS Post-ICU COVID-19 tool were used to describe identified domains of PICS in numbers and percentages. As appropriate, quantitative and continuous data were expressed as means with the standard deviation (SD) or as medians with interquartile ranges (IQRs). All data were analyzed using a statistical software package (SPSS Inc., version 24, Chicago, IL, USA). Additionally, we collected the free text of the reported consultations from the EHR and processed it anonymously into a text database for qualitative analysis. First, we checked if there was additional information reported about PICS symptoms according to the PICS framework (Table A1, Appendix C). Second, we collected data on reported patients’ needs, initiated interventions, and nursing problems. To support the data collection, we constructed an easy-to-use table that structured the data per patient. The data collection tables were pilot tested by three researchers (SH, AE, and JB) through independent data collection. Discrepancies in interpretation were resolved through discussion.

After pilot testing, two independent researchers read all of the notes from the structured telephone reports line by line (JB and SH). Both researchers (JB and SH) scrutinized the data by coding and extracting quotes [42]. In addition, a final category frame was made to structure all the relevant data into the three predetermined categories. The thematic matrix is attached in Appendix A. Both of the researchers categorized the data separately and discussed differences until a consensus was reached.

## 3. Results

In total, 49 post-ICU patients received an audio/video call in which the STS tool was used. The patient flow diagram shows the flow of patients (Figure 1); in three cases, data were missing because no information was reported. The first post-ICU patient who participated was admitted to the ICU on March 12, 2020, and the last patient was admitted on March 2, 2021. The median time between the ICU discharge and the interview was 37 days (IQR 24–54). The demographics and clinical characteristics of the included patients are presented in Table 1.

### 3.1. Reported PICS Symptoms

Table 2 details the scored symptoms of PICS with the Post-ICU COVID-19 tool. We described each element of PICS and combined the quantitative data from the structured tool and the qualitative data extracted from the text of the consultation reports of the ICU nurse practitioners as quotes (R#). After the structured questioning with the tool, there was also room to discuss other symptoms or to question symptoms in more depth. This additional information was reported as loose text. From this text, we organized the additional symptoms and describe them in Table A1 (Appendix C). Table A2 (Appendix C) shows an inventory of the professionals involved in the individual care plans of the post-ICU patients included.

#### 3.1.1. Physical Symptoms

All the patients experienced physical symptoms (*n* = 46). Over 60% of the patients experienced a loss of strength or condition. Fifty-four percent of patients reported the loss of muscle strength as “quite a lot”, and 59% reported the loss of condition as “quite a lot”. Dyspnea was scored as “very much” by 7% (*n* = 3) of the patients and as moderate by 26% (*n* = 12 patients). Fatigue was reported by 89% of patients (*n* = 39), and 22% (*n* = 10) of the patients reported this as intensely present.

The most frequent additional physical symptoms reported by the patients were dyspnea, poor condition, pain, and limited mobility. In particular, pain while breathing and fatigue were often mentioned as a limitation of daily function: 

*“In his own words, he sometimes ‘gasps for breath’”*.(R#7)

Patients experienced additional physical symptoms such as edema of the legs and sleeping problems. Sleeping problems were often mentioned, sometimes due to dyspnea, nocturnal urination, or psychological symptoms. If sleep problems were mentioned, a classification was lacking. 


*“I still have a lot of thoughts; I sleep an average of 2 to 3 h a night”.*
(R#15)

In addition, patients experienced weight loss and ICU-acquired weakness. 


*“Fine motor skills can still be improved, tingling in toes, colder hands are described”.*
(R#46)

#### 3.1.2. Cognitive Symptoms

Cognitive symptoms were less frequently reported; the inability to plan (13%, *n* = 6), the inability to multitask (15%; *n* = 7), and overstimulation (13%; *n* = 6) were reported as “quite a lot”. In addition, patients’ symptoms regarding cognitive dysfunction were reported, especially memory loss (9% reported this as “very much”) and concentration problems. Memory loss causes limitations in daily functioning and sometimes in the context of work.

*“Patient cannot carry out his work as a lawyer because of fatigue and memory loss”*.(R#3)

#### 3.1.3. Psychological Symptoms

All patients experienced psychological symptoms; most of the symptoms were signs of Post-Traumatic Stress Disorder (PTSD) and were reported as “quite a lot” in 17% of cases (*n* = 8). The most mentioned symptoms of anxiety were excessive worry and irritability. Patients expressed their concerns about COVID-19 and feared a recurrence of illness.

*“Patient expresses difficulties to deal with visitors because of fearing a recurrence of COVID-19”*.(R#14)

During the telephone consultations, emotions were often still present. For some patients, the conversation was still too tiring or had too much of an impact. Some patients explained a state of avoidance around memories of the ICU and described flashbacks, including nightmares. 

*“Patient becomes emotional several times—nightmares, thoughts of the delirium he experienced. He mentioned, he was tied up, wanted to take out the ventilator. He understands that this was necessary, but now he has terrible thoughts about it that continue to haunt him”*.(R#15)

#### 3.1.4. Caregivers’ Symptoms

In nine cases, caregivers participated in the video or telephone call. In 91.3% (*n* = 42) of cases, caregivers reported experiencing “not very many” symptoms. In four cases, the caregivers did report psychological symptoms. Most symptoms concerned fatigue and sleeping problems: *“The caregiver is more tired lately, sleeps worse, since last days” (R#5)*. When caregivers participated in the phone calls, they could explain which symptoms were present; in other cases, the patients themselves described the symptoms of their relative or caregiver. Fatigue, anxiety, flashbacks, and nightmares were experienced as symptoms. 

*“Anxious son, doing better, very afraid [for] his father’s health and the possibility of him getting a relapse of COVID”*.(R#24)


*“Husband sleeps only one hour a night, because he dreams of time in ICU”.*
(R#36)

In one case, relational problems were reported:

*“Patient experiences stress from memories of the ICU admission, and this manifested itself in problems in communication with his family members (informal caregivers)”*.(R#26)

### 3.2. Patients’ Needs

Based on the reported symptoms, the nurse practitioners detected patients’ needs based on their reported experience and symptoms. Patients especially mentioned the need for a continuity of care. This manifested itself primarily in the specific need for clear communication with their (primary) healthcare practitioners and clarity about follow-up at home. In addition, family caregivers expressed their need to be seen and heard in their role as informal caregivers. As a result, the ICU nurse specialist scheduled a follow-up meeting to listen to their experiences again as an aftercare measure. When healthcare practitioners were not involved, the ICU nurse specialist actively referred patients or caregivers to healthcare practitioners. In several cases, the ICU nurse specialist contacted the general practitioner or a medical specialist to improve the care plan—for instance, when symptoms had worsened or new symptoms had appeared. For some patients, the ICU nurse specialist could provide clarity about their admission into the ICU and provide information to meet their needs. Patients and family members always received an informal invitation to seek contact with the ICU nurse specialist in case of new problems. In some cases, the ICU nurse practitioners initiated extra follow-up calls for evaluating initiated advice and monitoring symptoms. Table A2, Appendix C shows an overview of the inventory of professionals already involved in the individual care plans of the post-ICU patients included.

## 4. Discussion

This study describes a quality improvement during the COVID-19 pandemic using the “STS Post-ICU COVID-19 tool”. We offered structured telephone follow-up care to post-ICU patients who were discharged from the hospital to the home within four weeks after hospital discharge. Forty-six post-ICU patients experienced physical symptoms, whereas cognitive and psychological symptoms were reported less. Almost two-thirds of the participating patients mentioned a loss of strength and condition. Family caregivers expressed the need for a continuity of care and experienced fewer symptoms.

Recent studies focused on physical function, reported fatigue, and muscle weakness as the most common symptoms after ICU admission with COVID-19 pneumonia [43]. We compared our study with a recent Dutch multicenter study that included 246 patients who were alive one year following ICU treatment for COVID-19 [44]. In this study, 74.3% reported physical symptoms, 26.2% reported mental symptoms, and 16.2% reported cognitive symptoms [44]. These findings are in line with our own. It is notable that our intervention was a self-report and was conducted shortly after hospital discharge (within 4 weeks). Another Dutch study including post-ICU patients without COVID-19 reported similar rates in terms of physical and cognitive symptoms. The rates of psychological symptoms were higher in this study [44]. Martillo et al. found a high prevalence (91%) of PICS elements in 45 COVID-19 post-ICU patients in a similar study with telephone follow-up after 1 month post-discharge. In this cohort, 87% reported physical impairments [32]. A study from Italy found reduced functional capacity in post-ICU patients with COVID-19 at 2 months post-hospital discharge [45]. The cardiopulmonary performance was presumably better in this study than in cohorts with other forms of ARDS. Carenzo et al. also highlighted the mildly reduced overall quality of life and a high proportion of PTSD symptoms at 6 months [45]. Halpin et al. (2020) implemented a similar rapid structured telephone tool, ordered by the International Classification of Functioning, Disability, and Health instead of the PICS classification we used [46]. Several studies reported physical symptoms such as fatigue, breathlessness, and psychological distress in COVID-19 survivors (*n* = 201) 14 and 20 days after hospital discharge [46,47]. In our study, we performed a follow-up at a median of 37 days (IQR 27-54), and our findings are in line with these aforementioned studies [46,47] 

A possible explanation for the high rate of physical symptoms is the higher proportion of ICU patients who received mechanical ventilation. A recent review showed that the number of ICU patients requiring mechanical ventilation during ICU admission increased from 20–40% in the period before COVID-19 to 63–87.3% during the COVID-19 pandemic [48]. Furthermore, the usual standards of care, such as the ABCDEF bundle, were not always practically feasible, which ideally would contribute to preventing PICS-F [49,50,51]. The symptoms of the family caregivers in our study were unexpectedly low. In nine cases, caregivers reported their concerns, and in the other cases, patients were asked to assess the symptoms of their caregiver; these assessments may have differed from the perceptions of the caregivers themselves. Nevertheless, the risk of suicide and self-harm has been associated with ICU admission when patients have pre-existent psychological conditions such as anxiety, depression, or PTSD [52]. This new insight supports the need for an individual assessment to determine the risk of developing symptoms of PICS and to initiate early interventions.

In our study, cognitive function scored low (“not”/“not often”), which is not in line with the current literature. The incidence of cognitive impairment after one year post-ICU was determined to be 43% [53]. In our study, patients were interviewed shortly after hospital discharge and were asked to self-assess their cognitive function. Early after discharge, the focus of most patients was on physical rehabilitation. Cognitive impairments may reveal themselves later; several studies have reported cognitive symptoms after a follow-up period of 2–156 months [53,54,55]. The self-assessment of cognitive function may be insufficient. Moreover, cognitive problems such as difficulty concentrating and multitasking often come to light when work and social obligations are resumed. However, the early onset of symptoms and the need for extra help could be detected and could be helpful in reducing long-term PICS complaints. 

The incidence of psychological symptoms was relatively low overall, except for the number of patients who reported PTSD symptoms (*n* = 8, 17%). A meta-analysis found that one-fifth of ICU survivors reported symptoms of PTSD after one year in the period before COVID-19. These numbers suggest that the psychological impact is high among survivors who have had “good” outcomes because they were discharged to home. We expect the numbers to be higher at a 6- or 12-month follow-up, as the impact on daily life becomes more apparent. However, recent studies report otherwise [44]. The caregivers in this study are underrepresented because they only participated in the consultations in a few cases. The few who reported their experience did describe psychological symptoms. Experts caution that the COVID-19 pandemic may have a greater impact on PICS-F numbers as family caregivers experience more awareness of the limitations of this crisis [56]. For family caregivers, a proactive follow-up call may have a positive impact in allowing them to share their experience and burden in order to access help early.

Although several studies have described the burden on ICU patients with COVID-19 and their families after hospital discharge, there is less evidence of initiated interventions. Before the COVID-19 pandemic, there was little evidence of interventions that could prevent long-term symptoms such as those caused by PICS [12]. 

This study describes a quality improvement in a single-center context and with a small sample size. Another limitation to mention is the fact that we only included patients discharged from the hospital to the home, which may not be representative of the whole population of ICU patients in this period. Almost one-third of COVID-19 patients admitted to Dutch ICUs died in the hospital [57]. The majority of the ICU patients admitted during this study period were transferred to a rehabilitation center or nursing home and received a structured rehabilitation care plan. For patients discharged to home, a structured care plan was lacking. Secondly, not all eligible patients received interventions due to limited staff capacity during the COVID-19 pandemic, and not all calls from patients were answered. In addition, a substantial group of ICU patients were transferred to another ICU because of hospital bed capacity problems during the pandemic. This may have led to selection bias. In addition, we know that having a loved one admitted to the ICU can have a major impact on family caregivers, especially during the COVID-19 pandemic. 

Nevertheless, family caregivers are underrepresented in this study, but they may have suffered more because the limited access to healthcare affected this group during the COVID-19 pandemic. This study substantiates that a structured, early inventory of PICS may potentially contribute to the proactive deployment of quality improvements. However, owing to a lack of time due to COVID-19, we did not perform an appropriate validation of the protocol. This tool may be beneficial for post-ICU patients and families in general in a non-pandemic setting. Further research is needed to validate and evaluate the effectiveness of structured tools to help professionals identify patients’ needs. In addition, cognitive function was measured by self-assessment; this may have resulted in reporting bias. Lastly, the effect of the coordinating role of nurse practitioners remains underrepresented in this study. In further research, we recommend focusing on the process of individual care planning during complex transitions after ICU admission. Before COVID-19, Dutch ICU aftercare was not structured or structural [8]. Whether aftercare should be structured or more individually based is a topic of debate. Therefore, an early inventory may provide guidance for individual care plans during the transition period from the hospital to the home.

## 5. Conclusions

To conclude, this quality improvement study shows that post-ICU patients and their family caregivers experienced symptoms of the PICS framework, identified at an early stage of post-discharge with the STS tool. Post-ICU patients diagnosed with COVID-19 pneumonia all reported symptoms in at least one of the three elements of the PICS framework. The most common symptom was physical burden in all patients, and symptoms of PTSD were also notable. We recommend individually planned early-onset rehabilitation and for more structured attention to be paid to family caregivers. Further research and follow-up are crucial, as COVID-19 is a new illness and post-discharge symptoms and long-term follow up are yet to be researched. 

## Figures and Tables

**Figure 1 ijerph-19-09689-f001:**
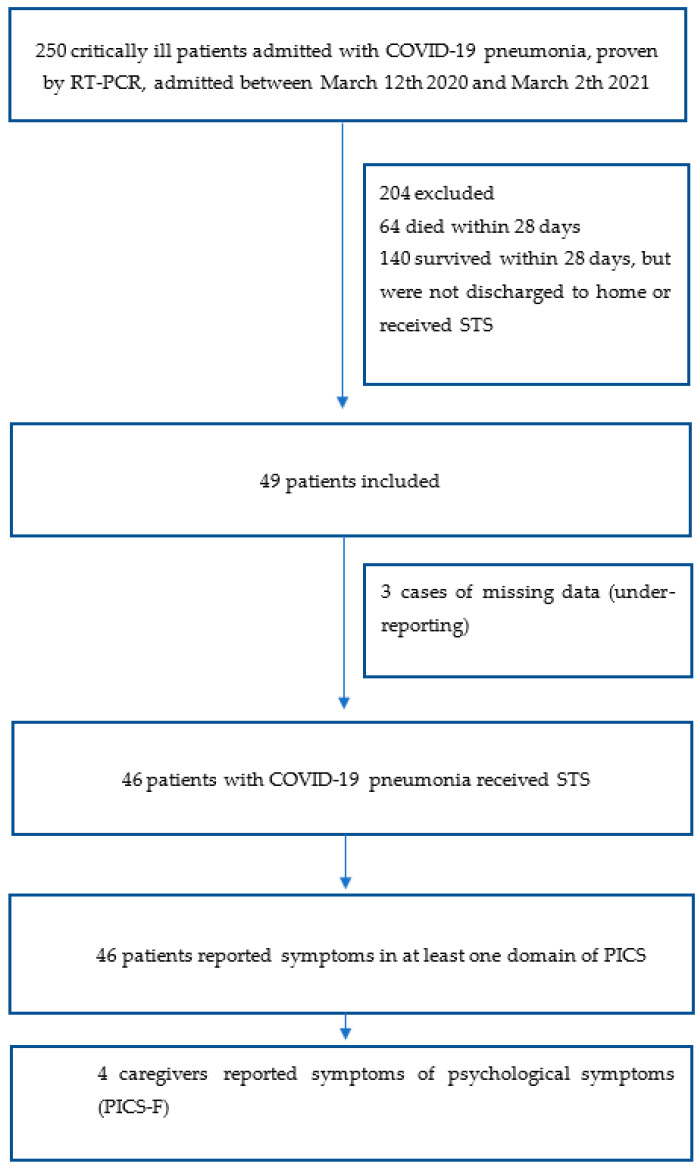
Patient flow diagram.

**Table 1 ijerph-19-09689-t001:** Baseline characteristics of the patients with COVID-19 pneumonia.

Received STS	Number of Patients
Age (years) (median, IQR)	62 (56–67)
Male gender (*n*, %)	35 (75%)
BMI (kg/m^2^) (median, IQR)	26.9 (24.6–31.7)
**Pre-existing comorbidities (*n*, %)**	
Obesity (BMI^3^ 30 kg/m^2^)	14 (30%)
Hypertension	13 (28%)
Congestive heart failure	4 (9%)
COPD	8 (17%)
Diabetes mellitus	9 (19%)
Cerebrovascular disease	3 (6%)
Malignancy	7 (15%)
Chronic renal disease	3 (6%)
Auto-immune disorder	5 (11%)
**Severity of illness**	
Sepsis-3, sepsis (*n*, %)	46 (100%)
Sepsis-3, septic shock (*n*, %)	6 (13%)
APACHE IV (points) (median, IQR)	47 (36–59)
SOFA (points) (median, IQR)	4 (2–6)
**ICU therapy during ICU stay (*n*, %)**	
Invasive mechanical ventilation	41 (89%)
HFNO only	5 (11%)
Vasoconstrictive agents	36 (78%)
Renal replacement therapy	4 (9%)
**ICU outcome**	
Duration of invasive mechanical ventilation (days) (median, IQR)	12 (6-22)
ICU LOS (days) (median, IQR)	12 (8-29)
Hospital LOS (days) (median, IQR)	21 (13–34)
28-day survival (N, %)	46 (100%)

Legends: All continuous data are presented as the median (interquartile range), and all categorical data are presented as a number (percentage). BMI: body mass index, COPD: chronic obstructive pulmonary disease, APACHE IV: Acute Physiology and Chronic Health Evaluation IV, SOFA: Sequential Organ Failure Assessment, HFNO: high-flow nasal oxygen, LOS: length of stay.

**Table 2 ijerph-19-09689-t002:** Elements of post-intensive care syndrome (PICS) [9] scored by the STS Post-ICU COVID-19 tool.

Physical Function, *n* = 46 Reported Physical Symptoms (Median 7.0, IQR 5–8.75)					
	Not	Not Very much	Quite a Lot	Very Much	Not Assessable
Loss of muscle strength	-	17 (37%)	25 (54%)	3 (7%)	
Loss of condition	-	11 (24%)	27 (59%)	6 (13%)	
Respiratory failure	11 (24%)	18 (39%)	12 (26%)	3 (7%)	
Fatigue	5 (11%)	10 (22%)	19 (41%)	10 (22%)	
Neuropathy	27 (59%)	11 (24%)	4 (9%)	-	
Cognitive function (median 4.0, IQR 0–7.75)					
Inability to plan	24 (52%)	7 (15%)	6 (13%)	-	8 (17.4%)
Memory loss	29 (63%)	13 (28%)	4 (9%)	-	-
Inability to concentrate	29 (63%)	10 (22%)	5 (11%)	-	1 (2%)
Inability to multitask	22 (48%)	9 (20%)	7 (15%)	1 (2%)	7 (15%)
Overstimulation	30 (65%)	10 (22%)	6 (13%)	-	-
Psychological burden (median 0.0, IQR 0.00–1.00)					
Feelings of anxiety	33 (72%)	10 (22%)	2 (4%)	1 (2%)	
Feelings of depression	40 (87%)	4 (9%)	-	2 (4%)	
Symptoms of PTSD	38 (83%)		8 (17%)		
Caregivers’ reported burden		42 (91.3%)	3 (6.5%)		1 (7.6%)

## Data Availability

The data informing the findings of this study are available from the corresponding author upon reasonable request.

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
