# Peer review of "A Quality Improvement Project to Support Post-Intensive Care Unit Patients with COVID-19: Structured Telephone Support"

_ijerph, 2022, doi:10.3390/ijerph19159689_

Round 1
Reviewer 1 Report
Thank you for the opportunity to review this paper which attempts to describe an intervention involving post - ICU discharge continuity of care delivered to COVID-19 patients through structured telephone support. The issue is very relevant considering that during a global health emergency such as that of COVID-19 pandemic highlighted the importance of overcoming physical restrictions imposed by protection measures and enhancing communication through telephone or other online devices between healthcare team, patents and families. Innovative interventions in these regards have been undertaken in many healthcare facilities across the world in an attempt to adapt to the emergency situation and respond to patients and relatives needs as best as possible (see for instance doi.org/10.1037/tra0000827; ).The importance of online communication between patients, families and healthcare staff has also been hghlighted in recent work as a crucial facilitator of continuity of care in a pandemic emergency context (see doi: 10.1016/j.iccn.2020.102893; doi: 10.1080/20008198.2021.1968141). The intervention that the authors try to describe is situated in this larger emerging literature and could provide an added value to it as it focuses on the importance of follow up and post-ICU care provided through telephone. However there are several problems with the way the current manuscript is structured.
- First, I did not find the introduction section anything on similar works on this topic, describing similar interventions (as cited above) which would largely increase the value of the current paper and situate it within a larger related literature.
- Secondly (and more importantly) it is not clear what kind of paper the authors are presenting, what is the study design? At times authors present quantitative data, other times qualitative excerpts from patient interviews, while the impression one gets from the title of this work is that of a description of an innovative intervention to improve quality of continuity of care during the pandemic. I would suggest the authors thoroughly revise the manuscript and re-submit it as a Letter to the Editor type of article where they narrate in a descriptive way the intervention in question and discuss implications that it may have for improving patient care, without presenting quantitative/or qualitative data that given the small sample would not add much value to the current paper.
- Lastly, the language needs reviewing as quite often there are sentences that that seem incomprehensible. For instance in the abstract one reads ..."the ICU nurse practitioners were able to check if individual care plans were optimal, clear, and if indicated, initiated disciplines to optimize further follow-up."
Author Response
Dear reviewer, thank you for your feedback and insightful comments and suggestions to improve the content of this EBQI paper. We hope your points of feedback correctly changed as we highlighted these with red colored text in the new version of the manuscript. Please see the attachment.

Reviewer 2 Report
Well-written and interesting manuscript. I note and suggest the following:
1. Introduction: (a) Update the number of COVID-19 cases admitted to ICUs to current
2. Methods: (a) STS has been explained well but was any formal process of validation of the protocol done? If not, then it must be listed as a limitation; (b) Ethical procedures good
3. Results: (a) Flow diagram helpful; (b) qualitative data presented well
4. Discussion: (a) Comparison to Dutch study good (lines 283-286 Reference 27) but the reviewer wonders if any such study is available globally for comparison. If so, it would be worthwhile including them; (b) References 28 & 29 seem to point the post COVID-19 symptoms being worse at 14-20 days but your study was conducted at a median of 37 days (IQR 27-54) so please explain this more; (c) Please elaborate on self assessment of cognitive functioning a s a limitation of your study
Overall, good effort and deserves publication.
Author Response

(The authors gave the same response as above.)

Round 2
Reviewer 1 Report
Whereas the authors have responded to some of the issues, I feel they have not carefully considered and reflected on my first comment which suggest them to place their study in a larger background of similar interventions undertaken around the world to improve not just patient outcomes, but also family and healthcare worker related ones.
In their response letter authors state that: "We assessed the suggested references to be valuable to enlarge the body of knowledge. We, therefore, made some additional changes in the introduction section.", but they provide no specific details about these changes and reflected references.
In lines 73-75 they have added in red the following sentence “Due to COVID-19 and visiting restrictions, virtual communication such as structured telephone support (STS) has been fully embraced as an intervention to keep the patient and family involved (13).” Reference 13 here seems to be completely unrelated to the inserted text in the manuscript (see reference below). I urge authors to more carefully examine this issue.
13. Feltner C, Jones CD, Cené CW, Zheng Z-J, Sueta CA, Coker-Schwimmer EJ, et al. Transitional care interventions to prevent readmissions for persons with heart failure: a systematic review and meta-analysis. Annals of internal medicine. 2014;160(11):774- 458.
Author Response
Thank you very much for your previous comments that helped us improve this manuscript. We understand there is a piece of important information missing in the introduction section. We revised the introduction section carefully and we refer to supportive literature. We highlighted this text changes within the revised manuscript of round 2, using blue-colored text. We hope the manuscript has been improved accordingly.